# Regulation of Cathelicidin Antimicrobial Peptide (CAMP) Gene Expression by TNFα and cfDNA in Adipocytes

**DOI:** 10.3390/ijms242115820

**Published:** 2023-10-31

**Authors:** Alexandra Höpfinger, Andreas Schmid, Leonie Schweitzer, Marissa Patz, Anja Weber, Andreas Schäffler, Thomas Karrasch

**Affiliations:** Department of Internal Medicine III, University of Giessen, 35392 Giessen, Germanyandreas.schaeffler@innere.med.uni-giessen.de (A.S.); thomas.karrasch@innere.med.uni-giessen.de (T.K.)

**Keywords:** TNFα, cfDNA, TLR (toll-like receptor) 9, cathelicidin anti-microbial peptide, adipocyte, adipose tissue, innate immunity

## Abstract

Understanding the complex interactions between metabolism and the immune system (“metaflammation”) is crucial for the identification of key immunomodulatory factors as potential therapeutic targets in obesity and in cardiovascular diseases. Cathelicidin antimicrobial peptide (CAMP) is an important factor of innate immunity and is expressed in adipocytes. CAMP, therefore, might play a role as an adipokine in metaflammation and adipose inflammation. TNFα, cell-free nucleic acids (cfDNA), and toll-like receptor (TLR) 9 are components of the innate immune system and are functionally active in adipose tissue. The aim of the present study was to investigate the impact of TNFα and cfDNA on CAMP expression in adipocytes. Since cfDNA acts as a physiological TLR9 agonist, we additionally investigated TLR9-mediated CAMP regulation in adipocytes and adipose tissue. CAMP gene expression in murine 3T3-L1 and human SGBS adipocytes and in murine and human adipose tissues was quantified by real-time PCR. Adipocyte inflammation was induced in vitro by TNFα and cfDNA stimulation. Serum CAMP concentrations in TLR9 knockout (KO) and in wildtype mice were quantified by ELISA. In primary adipocytes of wildtype and TLR9 KO mice, CAMP gene expression was quantified by real-time PCR. CAMP gene expression was considerably increased in 3T3-L1 and SGBS adipocytes during differentiation. TNFα significantly induced CAMP gene expression in mature adipocytes, which was effectively antagonized by inhibition of PI3K signaling. Cell-free nucleic acids (cfDNA) significantly impaired CAMP gene expression, whereas synthetic agonistic and antagonistic TLR9 ligands had no effect. CAMP and TLR9 gene expression were correlated positively in murine and human subcutaneous but not in intra-abdominal/visceral adipose tissues. Male TLR9 knockout mice exhibited lower systemic CAMP concentrations than wildtype mice. CAMP gene expression levels in primary adipocytes did not significantly differ between wildtype and TLR9 KO mice. These findings suggest a regulatory role of inflammatory mediators, such as TNFα and cfDNA, in adipocytic CAMP expression as a novel putative molecular mechanism in adipose tissue innate immunity.

## 1. Introduction

The worldwide prevalence of obesity and metabolic syndrome is continuously increasing [1,2,3]. Metabolic syndrome is one of the main risk factors for cardiovascular diseases [4] that are the number one cause of death in public health statistics [5]. Its underlying complex interactions have not been fully elucidated until now. In particular, chronic low-grade inflammatory state, adipose inflammation, and metaflammation in obesity are assumed to play a crucial role in the development of local and systemic insulin resistance, cardiovascular diseases, and impaired immune responses [4]. Recently published data argue for a role of adipose tissue innate immunity in regulating the molecular interface of metabolism and inflammation [6,7]. 

Cathelicidin antimicrobial peptide (CAMP) represents an immunomodulatory peptide mainly secreted by immune cells such as monocytes, macrophages, and lymphocytes [8] with a crucial role in the innate immune system. CAMP enhances monocyte [9] and granulocyte [10] infiltration at sites of local inflammation by chemoattractant action and alters the immune response in the course of infection via toll-like receptor (TLR) modulation [11]. In bacteria, CAMP causes a loss of membrane integrity by increased permeability, leading to the death of microbial cells [12].

In 2015, for the first time, Zhang et al. described CAMP as being produced by adipocytes [13], thereby introducing CAMP as a novel adipokine. During subcutaneous Gram-positive infection with Staphylococcus aureus, CAMP secreted by adipocytes inhibits bacterial growth and is an important protective component of subdermal adipose tissue against bacterial infection [13]. In mice, diet-induced obesity results in the loss of a dermal pool of preadipocytes and thus inhibits the capacity to initiate reactive adipogenesis and to express CAMP [14]. In humans, CAMP gene expression is higher in subcutaneous than in visceral adipose tissues [15]. However, further data on the precise function and regulation of CAMP in adipocytes are sparse. 

In 2021, we elucidated regulatory mechanisms of CAMP expression in murine 3T3-L1 adipocytes [16]. Activation of TLR2 and TLR4 by specific ligands (MALP2, lipopolysaccharides (LPS)) up-regulates adipocyte CAMP expression involving classical signal transduction elements such as NF-ĸB, PI3K and STAT3 [16]. This TLR-mediated proinflammatory activation of CAMP expression can be modified by immunomodulatory adipokines such as C1q/TNF-related protein-3 (CTRP-3) [17]. Furthermore, immuno-metabolic factors, such as bile acids, glucose, insulin, and incretins, are able to modulate CAMP expression in adipocytes in vitro [15].

In obesity, the death of adipocytes leads to an increase of cell-free nucleic acids (cfDNA) in adipose tissue and plasma [18], and adipose tissue-resident macrophages are involved in the recognition of these elevated cfDNA levels via TLR9 [18]. TLR9 is a well-characterized intracellular receptor [19] recognizing non-methylated CpG DNA motifs derived from bacteria [20], viral double-stranded DNA (dsDNA) [21], and cfDNA as a signal for severe tissue damage inside the organism [18]. Activation of TLR9 leads to an increased production of proinflammatory cytokines [22], and TLR9 signaling in macrophages is involved in obesity-induced insulin resistance [23]. Of note, Nakagawa et al. demonstrated that CAMP is required for normal TLR9 function in dendritic cells and macrophages [24]. Interestingly, adipocytes themselves express functionally active TLR9 [25]. Whether CAMP plays an additional role in TLR9 modulation in adipocytes has not been elucidated so far. A possible interaction between CAMP expression, TNFα, and cfDNA in adipocytes and adipose tissues might provide a novel molecular interface of obesity-related inflammation with adipocyte function. 

Therefore, the primary aim of the present study was to investigate:The underlying mechanisms involved in TNFα-mediated regulation of CAMP expression in adipocytes with respect to components of established proinflammatory signaling pathways;The potential effects of cfDNA, representing an inflammatory mediator in adipose tissue and a physiological TLR9-agonist, on CAMP gene expression in murine 3T3-L1 adipocytes;Circulating CAMP concentrations as well as local CAMP gene expression in adipose tissue and primary adipocytes of TLR9 knockout mice compared to wildtype mice.

## 2. Results

### 2.1. CAMP Gene Expression Is Induced during Differentiation in Murine 3T3-L1 Adipocytes and in Human SGBS Cells

During the differentiation process, 3T3-L1 adipocytes phenotypically turn from fibroblastic preadipocytes into mature adipocytes with multiple lipid vacuoles within 8 days. The pattern of adipokine expression and secretion varies depending on the stage of differentiation. In early stage 3T3-L1 preadipocytes, at days 0 and 3 of differentiation, no significant CAMP gene expression was detected. In later stages, at day 6 and in mature adipocytes (day 8), a strong and significant increase of CAMP expression was observed (Figure 1C). Human SGBS preadipocytes were differentiated over 14 days into mature adipocytes. In early preadipocytes, there was no significant CAMP expression. During differentiation, CAMP expression increased strongly until day 8, remaining at an elevated level during final adipocyte maturation (Figure 1F). 

### 2.2. TNFα-Induced CAMP Gene Expression Can Be Antagonized by Inhibitors of Intracellular Signal Transduction Pathways

In 3T3-L1 adipocytes, a state of inflammation was induced by stimulation with TNFα. Treatment with doses of 2 ng/mL TNFα (*p* = 0.005) and 10 ng/mL TNFα (*p* < 0.001) significantly increased CAMP gene expression (Figure 2A). Inhibition of the PI3K (by 5 µM LY294002) signal transduction pathway effectively antagonized TNFα-induced CAMP gene expression (*p* = 0.032) (Figure 2B), whereas inhibition of NF-κB (5 µM BAY11-7085), STAT3 (50 µM S3I-201), MAPK (5 µM SB239063), or MEK-1/2 (5 µM U0126) signal transduction pathways did not antagonize TNFα-induced CAMP expression (Figure 2B,C). In contrast, CAMP mRNA levels were considerably elevated by 50 µM S3I-201 and 5 µM U0126 treatment, in addition to the TNFα-induced effect (Figure 2C). Basal CAMP mRNA levels were not significantly affected by inhibition of any of these signal transduction pathways (Figure 2D).

### 2.3. Cell-Free Nucleic Acids (cfDNA) Impair CAMP Expression in 3T3-L1 Adipocytes

Adipocytes release cfDNA as a response to inflammatory stress [18] and cfDNA acts as an endogenous ligand for TLR9 in immune cells [26]. In the present study, we investigated whether cfDNA from adipocytes has an impact on CAMP expression in adipocytes. In 3T3-L1 adipocytes, cfDNA reduced CAMP expression with an effective dosage of 1 µg/mL cfDNA (*p* = 0.002), whereas 100 ng/mL proved to be ineffective (Figure 3). 

### 2.4. CAMP and TLR9 mRNA Levels Are Positively Correlated in Human Subcutaneous Adipose Tissue 

cfDNA is increased in the adipose tissue and plasma of obese patients [18]. Since cfDNA represents a physiological TLR9 agonist [27] and cfDNA inhibited CAMP gene expression in adipocytes in vitro (Figure 3), gene expression levels of CAMP and TLR9 were analyzed using quantitative real-time PCR in subcutaneous and visceral adipose tissue specimens obtained from obese individuals undergoing bariatric surgery. Interestingly, TLR9 and CAMP gene expression levels in subcutaneous adipose tissue were positively correlated (*rho* = +0.282; *p* = 0.008; n = 88) (Figure 4A). No correlation of TLR9 and CAMP expression was detected in human visceral adipose tissue (*rho* = +0.034; *p* = 0.747; n = 90). Furthermore, CAMP mRNA quantities in human visceral and subcutaneous adipose tissue compartments correlated positively (*rho* = +0.224; *p* = 0.005; n = 157) (Figure 4B).

### 2.5. CAMP and TLR9 Gene Expression Levels Are Positively Correlated in Murine Subcutaneous Adipose Tissue 

It is well known that subcutaneous and intra-abdominal adipose tissues differ in the numerical ratio of adipocytes to stromal vascular cells as well as in functional characteristics [28]. Therefore, we additionally investigated both adipose tissue compartments separately in murine samples. Subcutaneous and intra-abdominal adipose tissue specimens were resected from wildtype C57BL/6J mice. Similar to human adipose tissue samples, CAMP and TLR9 gene expression levels in subcutaneous adipose tissue were found to be positively correlated (rho = +0.657; *p* = 0.011; n = 14) (Figure 5). In contrast, there was no significant correlation of TLR9 and CAMP mRNA levels in intra-abdominal adipose tissue (rho = −0.178; *p* = 0.543; n = 14). 

### 2.6. TLR9 Deficiency Is Associated with Reduced Systemic CAMP Serum Concentrations in Male Mice

To further investigate a possible interrelation between CAMP and TLR9 expression levels, we analyzed blood serum samples from wildtype mice and TLR9 KO mice. Serum samples were collected at the age of 5–8 months and circulating CAMP levels were quantified via ELISA. Of note, CAMP serum concentrations were significantly lower (*p* = 0.011) in male TLR9 KO mice when compared to wildtype mice (WT: 6122.70 ± 1962.79 pg/mL vs. TLR9 KO: 4445.01 ± 826.42 pg/mL) (Figure 6A). This difference between genotypes was not observed in female mice (WT: 3735.38 ± 973.51 pg/mL vs. TLR9 KO: 3937.66 ± 1847.37 pg/mL) (Figure 6B). 

### 2.7. Synthetic TLR9 Ligands Do Not Modify CAMP Gene Expression in Murine Adipocytes

In immune cells, the activation of TLR9 generally leads to an increased production of proinflammatory cytokines [22]. Oligodeoxynucleotides (ODNs) are established synthetic ligands of TLR9 [29]: ODN2087 is a commonly used TLR7/TLR9 inhibitor, whereas ODNA, ODNB, and ODNC represent agonists inducing TLR9 activity [25]. Therefore, we investigated whether ODNA, ODNB, ODNC, and ODN2087 modify basal CAMP gene expression in murine adipocytes in vitro (Figure 7). Doses of 5 µg/mL and 20 µg/mL ODNA, 5 µg/mL, 20 µg/mL ODNB, 10 µg/mL and 20 µg/mL ODN C, and 20 µg/mL ODN2087 were applied in murine 3T3L-1 adipocytes. However, none of the tested ODNs significantly affected adipocyte CAMP gene expression (Figure 7).

### 2.8. Knockout of TLR9 Does Not Affect CAMP Gene Expression in Murine Primary Subcutaneous and Intra-Abdominal Adipocytes

We detected significant correlations of CAMP and TLR9 gene expression in human and murine subcutaneous adipose tissue, and TLR9 knockout in male mice resulted in reduced CAMP serum levels. However, synthetic TLR9 ligands—as opposed to cfDNA—did not modify CAMP gene expression in murine adipocytes in vitro. In order to further investigate the impact of TLR9 on CAMP expression in adipocytes, we tested the cellular effect of TLR9 knockout on CAMP gene expression in primary adipocytes isolated from wildtype mice and TLR9 KO mice. Due to the diverging constitution of subcutaneous and intra-abdominal adipose tissue, we evaluated adipocytes from both compartments separately (Figure 8A,B). Of note, we detected no difference in CAMP gene expression in subcutaneous (Figure 8A) and intra-abdominal (Figure 8B) adipocytes of wildtype and TLR9 knockout mice.

## 3. Discussion

During adipocyte differentiation, the expression and secretion of various adipokines increase [30]. Zhang et al. reported that CAMP gene expression in mature murine 3T3-L1 adipocytes is increased when compared to preadipocytes [13], which was confirmed by our group [31]. In the experiments presented here, we could further demonstrate a strong and persistent induction of CAMP gene expression during differentiation of human SGBS preadipocytes into mature adipocytes, a finding comparable to the expression of CAMP during in vitro differentiation of primary human preadipocytes [13]. SGBS adipocytes [32] therefore might represent an applicable cell culture model for the analysis of CAMP regulation in human adipocytes, improving the translational potential of in vitro findings for future clinical and therapeutical applications. 

The capacity of host defense in subcutaneous bacterial infection depends on metabolic factors, which is demonstrated by impaired host defense in obesity [14]. Importantly, adipocyte-derived CAMP plays a crucial role in the local host defense of subcutaneous adipose tissue [13]. As an antimicrobial peptide, CAMP is part of the innate immune system. The regulation of CAMP in adipocytes by other factors of the innate immune system is unknown so far. The proinflammatory cytokine TNFα is expressed in adipose tissue and affects adipose tissue-resident macrophage activation [33]. In obesity, increased levels of TNFα are observed, and TNFα levels correlate with the extent of adiposity and associated insulin resistance [33]. Remarkably, we found that TNFα significantly induces CAMP gene expression in 3T3-L1 adipocytes. This inflammatory elevation of CAMP expression was antagonized by the inhibition of PI3K signaling. Of note, pharmacological antagonists of other classical signal transduction pathways exerted either no (NF-κB, MAPK) or even contrary effects (STAT3, MEK-1/2). Basal, non-stimulated CAMP gene expression levels were not affected by the inhibition of these signal transduction pathways, suggesting a specific TNFα-/inflammation-related mechanism.

In obesity, an increased quantity of visceral adipose tissue and adipocyte hypertrophy are associated with systemic low-grade inflammation, representing a key aspect of the metabolic syndrome [6]. Macrophage infiltration into visceral adipose tissue induces local inflammatory stress in adipocytes [18]. Subsequent adipocyte degeneration results in the release of nucleic acids (cell-free DNA, cfDNA) [18], presumably exerting paracrine effects on adipocytes [34]. In obesity, levels of cfDNA in circulating blood are increased and positively correlated with the quantity of visceral adipose tissue [18]. cfDNA has even been suggested as a biomarker for obesity [35]. Local cfDNA acts as an endogenous ligand of TLR9 on immune cells leading to an increased recruitment of these cells to the site of inflammation [26]. Of note, CAMP is required for normal TLR9 function in dendritic cells and macrophages [24]. Our group demonstrated previously that there is significant and functional TLR9 expression in adipocytes [25]. The activation or inhibition of TLR9 activity affects the pattern of adipokine secretion in 3T3-L1 adipocytes in vitro [25]. We therefore aimed to investigate whether adipocyte CAMP gene expression is modulated by cfDNA and if TLR9 is involved in this context.

Interestingly, stimulation with cfDNA significantly reduced CAMP gene expression, suggesting that cfDNA represents a yet unknown paracrine factor in adipose tissue inhibiting CAMP expression in adipocytes. As CAMP gene expression was regulated by the physiological TLR9 agonist cfDNA, we investigated a possible association between CAMP and TLR9 expression in vivo. Considering the differing cellular composition and function of subcutaneous and intra-abdominal adipose tissues, we comparatively investigated these compartments in mice and in human individuals. In subcutaneous adipose tissue of wildtype mice, TLR9 and CAMP mRNA levels were positively correlated, whereas no significant correlation of these parameters was observed in intra-abdominal adipose tissue. In accordance with our findings in mice, TLR9 and CAMP expression were positively correlated in human subcutaneous but not in visceral adipose tissue. Thus, our data indicate a putatively compartment-specific association of TLR9 and CAMP expression in subcutaneous fat. Future studies elaborating on these data should focus on primary adipocytes and immune cells residing in adipose tissue—such as macrophages and T cells—in order to elucidate which cell types predominantly contribute to the observed correlation, and also whether the differences between subcutaneous and intra-abdominal/visceral adipose tissue might depend on the diverging cellular composition of both compartments.

Additionally, CAMP serum levels were measured in C57BL/6J wildtype and TLR9 knockout mice. Systemic CAMP concentrations were significantly lower in TLR9-deficient mice when compared to wildtype animals. Of note, this difference between genotypes was exclusively observed in male but not in female mice. This unexpected sexual dimorphism needs to be further elucidated, bearing in mind that testosterone or estradiol did not exert a significant impact on CAMP expression in adipocytes in previous experiments in vitro [15]. In accordance with the present findings in mice, male patients in this previous study exhibited higher serum concentrations and subcutaneous adipose tissue mRNA levels of CAMP when compared to women [15]. Taken together, these data strongly argue for a sexual dimorphism in adipose tissue CAMP gene expression that might also affect circulating CAMP levels. With the underlying regulatory processes remaining to be elucidated, future studies should address this important issue at the molecular level of sexual hormones, potentially affecting the transcriptional and/or secretory regulation of CAMP in more detail. In particular, future experiments should investigate CAMP gene expression in primary murine adipocytes isolated from males versus females to further elaborate on a potential sexual dimorphism and on a potential impact of impaired adipocyte differentiation.

Since we were able to demonstrate a strong association between the gene expression of CAMP and TLR9 in (subcutaneous) adipose tissue and on systemic levels in vivo, we asked if this association might be mediated by TLR9-dependent CAMP regulation in adipocytes. Thus, we investigated the effect of synthetic TLR9 ligands on CAMP gene expression in adipocytes in vitro. Representing well-established TLR9 ligands, the oligodeoxynucleotides ODN1585 (ODNA), ODN1826 (ODNB), ODN2395 (ODNC), and ODN2087 are known to modify immune responses by modulating TLR9 activity [36]. Of note, CAMP gene expression was affected neither by ODNA, ODNB, ODNC, nor by ODN2087 in adipocytes. Additionally, we analyzed CAMP gene expression in primary adipocytes from wildtype and TLR9 knockout mice. Importantly, we observed no significant difference between CAMP gene expression levels in primary adipocytes from wildtype mice as compared to TLR9 knockout mice, neither for subcutaneous nor intra-abdominal adipose tissue. Together, these findings suggest that—although cfDNA impairs CAMP gene expression in adipocytes in vitro—the association of CAMP and TLR9 expression levels in adipose tissues presumably mainly depends on other cell types that are different from adipocytes, like monocytes or macrophages in the stromal vascular fraction of adipose tissue, which are a known source of CAMP [37]. Furthermore, the observed inhibitory effect of cfDNA on CAMP gene expression in adipocytes might be mediated via additional receptors, for example TLR7, as cfDNA is a putative TLR7 agonist as well [34]. Future studies on the putative regulatory interactions of CAMP and TLR7 in adipocytes are necessary to verify this hypothesis and to further elucidate the mechanisms underlying CAMP expression in adipose tissue. 

Overall, we focused on a gene expression analysis of CAMP in adipocytes and adipose tissue in order to gain a basic insight into the regulatory relations between the inflammatory factors of innate immunity and CAMP. The predominant analysis on the level of transcriptional regulation represents a limitation concerning the physiological conclusions drawn from the present work. Therefore, future research elaborating on the provided data should apply immunological techniques such as Western blot and immunocytochemistry in order to analyze CAMP regulation in adipocytes and adipose tissue at the protein level and to confirm and further elucidate the biological relevance of these processes.

## 4. Materials and Methods

### 4.1. 3T3-L1 Cell Culture and Stimulation Experiments

Murine 3T3-L1 fibroblasts [38] were cultured and differentiated into mature adipocytes as described previously [39]. Briefly, cells were cultured at 37 °C and 5% CO_2_ in Dulbecco’s Modified Eagle Medium (Biochrom AG, Berlin, Germany) supplemented with 10% newborn calf serum (Sigma-Aldrich, Deisenhofen, Germany) and were differentiated into adipocytes in DMEM/F12/glutamate medium (Lonza, Basel, Switzerland) supplemented with 20 µM 3-isobutyl-methyl-xanthine (Serva, Heidelberg, Germany), 1 µM corticosterone, 100 nM insulin, 200 µM ascorbate, 2 µg/mL transferrin, 5% fetal calf serum (FCS, Sigma-Aldrich, Deisenhofen, Germany), 1 µM biotin, 17 µM pantothenic acid (all from Sigma Aldrich, Deisenhofen Germany), and 300 µg/mL Pedersen-fetuin (MP Biomedicals, Illkirch, France) [40,41]. A differentiation protocol reported in the literature [38,42,43,44,45] was used with slight modifications. Visual control of the cellular phenotype and of lipid accumulation by light-microscopy was performed during all stages of the differentiation process.

Mature adipocytes were incubated under serum-free conditions prior to stimulation experiments. TLR9 agonistic oligodeoxynucleotide (ODN) ODN1585 (referred to as ODNA), ODN1826 (ODNB), ODN2395 (ODNC), and inhibitory ODN2087 were purchased from Invivogen (San Diego, CA, USA) and were dissolved in H2O under sterile conditions. ODNA (5 µg/mL, 20 µg/mL), ODNB (5 µg/mL, 20 µg/mL), ODNC (10 µg/mL, 20 µg/mL), and ODN2087 (20 µg/mL) were applied in two separate overnight (18 h) stimulation experiments. All applied doses were determined following the manufacturer’s recommendations for effective dosage as well as by previous preliminary testing for potential cytotoxicity in adipocyte culture. Cell-free nucleic acids (referred to as cfDNA) were isolated from 3T3-L1 adipocytes as described previously [34] and below, and were applied in doses of 100 ng/mL and 1 µg/mL. As an inflammatory stimulus, TNFα was applied at a dosage of 2 ng/mL and 10 ng/mL. Furthermore, co-stimulation experiments were performed with 10 ng/mL TNFα and inhibitors of different signal transduction pathways (NF-κB inhibitor BAY-11 (5 mM), STAT3 inhibitor S3I-201 (50 mM), selective MAPK inhibitor SB239063 (5 mM), MEK-1/-2 inhibitor U0126 (5 mM), and phosphatidylinositol 3-kinase (PI3K) inhibitor LY294002 (5 mM), all purchased from Merck). The dosage was applied as described in our previous study [16].

All applied stimulatory doses were within the concentration range recommended by the manufacturer and had been determined either by preliminary tests or previous experiments in adipocyte culture with respect to dose effects and cell viability. Furthermore, LDH (lactate dehydrogenase) concentration was measured in the cell supernatants (Cytotoxicity Detection Kit, Roche, Mannheim, Germany) of all experiments in order to exclude any unintended cytotoxic effects.

### 4.2. SGBS Cell Culture 

The Simpson–Golabi–Behmel syndrome (SGBS) preadipocyte cell strain represents primary human cells originated from adipose tissue specimens of a patient suffering from SGBS [32]. They were kindly provided by Prof. Martin Wabitsch (University of Ulm, Germany). The cells were differentiated into mature adipocytes within 14 days of culture following the provider’s established protocol [32]. SGBS preadipocytes were cultured in DMEM/F12 (1:1) (Invitrogen, Darmstadt, Germany) supplemented with 10% FCS (all purchased from Invitrogen, Darmstadt, Germany). Differentiation into mature adipocytes was induced at confluence. After 3 washing steps with phosphate-buffered saline (PBS), cells were cultured in serum-free medium supplemented with 0.01 g/mL transferrin, 20 nM insulin, 0.2 nM triiodothyronine, and 100 nM cortisol (all purchased from Sigma-Aldrich, Deisenhofen, Germany). During the initial 4 days of differentiation, 2 μM rosiglitazone (BRL 49653) (Cayman, Tallinn, Estonia), 250 μM isobutylmethylxanthine (IBMX), and 25 nM dexamethasone (all purchased from Sigma-Aldrich, Deisenhofen, Germany) were also added to the medium. The culture medium was replaced every third or fourth day. Adipocyte differentiation was completed after 14 days. The characteristic adipocyte morphology was visually controlled by light microscopy.

### 4.3. Preparation of Cell-Free Nucleic Acids from 3T3-L1 Adipocytes

Prior to isolation of cell-free nucleic acids (cfDNA) using the QIAamp^®^ DNA Micro Kit according to the manufacturer’s instructions (Qiagen, Hilden, Germany), murine 3T3-L1 adipocytes were exposed to inflammatory stress induced by stimulation with 50 ng/mL tumor necrosis factor α (TNFα) (Biomol, Hamburg, Germany). After 18 h, adipocytes were harvested and cfDNA was isolated from cell lysates as described earlier [34]. Differing doses of cfDNA (100 ng/mL; 1 µg/mL) were applied for overnight (18 h) stimulation experiments in 3T3-L1, while treatment with solvent control containing no cfDNA served as a control setting.

### 4.4. Human Visceral and Subcutaneous Adipose Tissues for Gene Expression Analysis

Human adipose tissue specimens were obtained from the ROBS (Research in Obesity and Bariatric Surgery) study [46], an open-label, non-randomized, monocentric, prospective, and observational study of patients undergoing either bariatric surgery or dietary intervention at the University Hospital of Giessen, Germany. Subcutaneous and visceral adipose tissues were obtained intra-surgically from patients receiving bariatric surgery (gastric sleeve or Roux-en-Y gastric bypass). Detailed information about the ROBS study cohort was published previously and can be drawn from the literature [46]. The study was approved by the local ethical committee at the University of Giessen, Germany (file: AZ 101/14). All patients gave informed consent for their participation in the study. Data anonymization and privacy policy were accurately applied.

### 4.5. Quantification of CAMP Gene Expression in Murine Intra-Abdominal and Subcutaneous Adipose Tissues and Primary Adipocytes

Intra-abdominal and subcutaneous adipose tissue compartments were resected from C57BL/6J-Tlr9^M7Btlr^/Mmjax mice (TLR9-KO) [25] and wildtype C57BL/6J mice (WT) that were bred under standard conditions and fed normal chow (age 5–8 months). Animals were euthanized for the sampling of serum, subcutaneous and intra-abdominal adipose tissue, and primary adipocytes conformably to the German animal protection law (§4 Abs. 3 *Tierschutzgesetz*). A specific announcement was made at the local ethical committee (*Regierungspraesidium* Giessen; internal registration number: 710_M) that was subsequently approved. 

Primary adipocytes from intra-abdominal and subcutaneous adipose tissue were isolated as described earlier [34]. Briefly, small portions of the intra-abdominal and subcutaneous adipose tissue from C57BL/6J-Tlr9^M7Btlr^/Mmjax mice (TLR9-KO) [25] and wildtype C57BL/6J mice (WT) were digested with 0.225 U/mL collagenase NB 6 (#17458, SERVA Electrophoresis; Heidelberg, Germany) for 30–60 min at 37 °C. In a subsequent centrifugation step (200 rcf, 10 min, 4 °C), the adipocytes were separated from the stromal vascular cells (SVC) and later transferred into TRIzol^®^-Reagent (Life Technologies GmbH, Darmstadt, Germany) for mRNA isolation, as described below (Section 4.6). The purity of the adipocytes versus SVC after our isolation procedure was assessed by real-time PCR, applying adiponectin as a biomarker for the adipocyte fraction and CD45 (indicating leukocyte lineage) for the stroma-vascular cell fraction [34].

### 4.6. Isolation of mRNA and Quantitative Real-Time PCR Analysis of CAMP Gene Expression in Murine and Human Adipose Tissues and Adipocytes

Total mRNA was isolated from frozen human and murine total adipose tissues and from 3T3-L1, SGBS, and primary adipocytes following a protocol described previously [39]. Tissues were homogenized in TRIzol^®^-Reagent (Life Technologies GmbH, Darmstadt, Germany) in combination with gentleMACS dissociator and M-tubes (Miltenyi Biotec GmbH, Bergisch Gladbach, Germany) for dissociation and RNA was isolated applying RNeasy^®^ Mini Kit (Qiagen, Hilden, Germany) including DNase (RNase-Free DNase Set, Qiagen, Hilden, Germany).

For the gene expression analysis, reverse transcription of RNA (QuantiTect Reverse Transcription Kit from Qiagen, Hilden, Germany) was performed in order to generate corresponding cDNA for real-time PCR (RT-PCR) (iTaq Universal SYBR Green Supermix, CFX Connect RT-PCR system; Bio-Rad, Munich, Germany). Expression levels of the target gene CAMP were normalized to the gene expression of glyceraldehyde-3-phosphate dehydrogenase (GAPDH) as a house-keeping gene applying the ΔΔCT method. The following primer-pairs were used:Murine CAMP:5′-CCCAAGTCTGTGAGGTTCCG-3′/5′-GTGCACCAGGCTCGTTACA-3′Murine GAPDH:5′-TGTCCGTCGTGGATCTGAC-3′/5′-AGGGAGATGCTCAGTGTTGG-3′Murine TLR9:5′-CATCTCCCAACATGGTTCTCC-3′/5′-GCAGAGAAACGGGGTACAGA-3′Human CAMP:5′-TAGATGGCATCAACCAGCGG-3′/5′-CTGGGTCCCCATCCATCGT-3′Human GAPDH:5′-GAGTCCACTGGCGTCTTCAC-3′/5′-CCAGGGGTGCTAAGCAGTT-3′Human TLR9:5′-CCCCCAGCATGGGTTTCT-3′/5′-TGGAGCTCACAGGGTAGGAA-3′

All oligonucleotides used were purchased from Metabion, Martinsried, Germany.

### 4.7. Quantification of CAMP Concentrations in Murine Blood Serum

Healthy C57BL/6J (WT) and C57BL/6J-Tlr9^M7Btlr^/Mmjax mice (TLR9-KO) were bred under standard conditions being fed a chow diet *ad libitum* until euthanasia for organ and tissue resection and blood serum collection. Concentrations of CAMP in blood serum from C57BL/6J-Tlr9^M7Btlr^/Mmjax mice (TLR9-KO) and wildtype C57BL/6J mice (WT) (age 5–8 months) were measured in technical duplicates by enzyme-linked immunosorbent assay (ELISA) (kit purchased from Abbexa Ltd., Cambridge, UK) and are expressed as means ± standard deviation in the text and displayed as box plots in the figures. The test range of the applied ELISA kit was 1.56–100 ng/mL. All measurements exceeding an intra-duplicate variance of 20% were repeated. Animal studies were performed at the University of Giessen, Germany, and were approved by the local government agency.

### 4.8. Statistical Analysis

For explorative data analysis, a statistical software package (SPSS 28.0) was used. Non-parametric numerical parameters were analyzed using the Mann–Whitney U-test (for 2 unrelated samples) or the Kruskal–Wallis test (>2 unrelated samples). A correlation analysis was performed by using the Spearman-rho test. *p* values below 0.05 (two-tailed) were considered statistically significant. In the figures, the data are presented as box plots, with the box representing the second and third quartile of values, the whiskers giving the inter quartile range, and statistical outliers outside inter quartile range being indicated by dots and stars.

## 5. Conclusions

As a part of the innate immune system, adipocytes, as well as adipose tissue as a whole organ, play a crucial role in immune-metabolism (“metaflammation”, “adipose inflammation”) [47]. Here, we investigate and discuss a novel aspect of the inflammation-mediated regulation of CAMP in adipocytes and adipose tissues. As visualized in Figure 9, our results implicate that CAMP expression in adipocytes is induced by TNFα involving the PI3K pathway. Furthermore, the physiological TLR7/TLR9 ligand cell-free nucleic acids (cfDNA) impairs CAMP gene expression in adipocytes. Further studies are necessary in order to investigate the molecular mechanisms and functional consequences of this novel relation within innate immunity and metaflammation in more detail. Most importantly, TNFα, CAMP, and adipose tissue-derived cfDNA might provide potential molecular targets in future anti-inflammatory drug therapies addressing metabolic inflammation and concomitant morbidities.

## Figures and Tables

**Figure 1 ijms-24-15820-f001:**
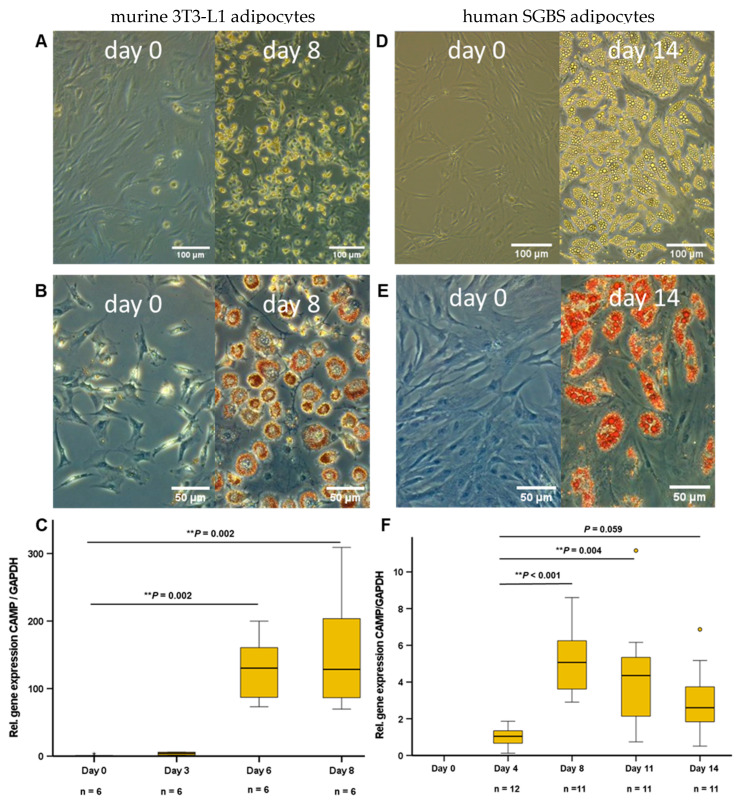
CAMP gene expression is induced during differentiation in murine 3T3-L1 adipocytes (**C**) and in human SGBS cells (**F**). In 3T3-L1 preadipocytes, no significant CAMP expression is detectable. At day 6 and in mature adipocytes, a considerable increase of CAMP mRNA levels is observed (**C**). CAMP expression significantly increases during differentiation of SGBS cells (**F**). Light microscopy of naïve (**A**,**D**) and Oil Red O stained (**B**,**E**) adipocytes is shown. During differentiation, 3T3-L1 fibroblasts on day 0 mature to adipocytes on day 8 (**A**,**B**). SGSB preadipocytes on day 0 mature to adipocytes on day 14 (**D**,**E**). CAMP expression was investigated by quantitative real-time PCR. The Kruskal–Wallis test was applied for calculation of *p* values and statistical significance (*p* < 0.05). N = 6–12 wells were investigated per experimental setting. Gene expression levels are given in relative gene expression as compared to day 0 in 3T3-L1 and day 4 in SGBS cells. (** *p* < 0.01).

**Figure 2 ijms-24-15820-f002:**
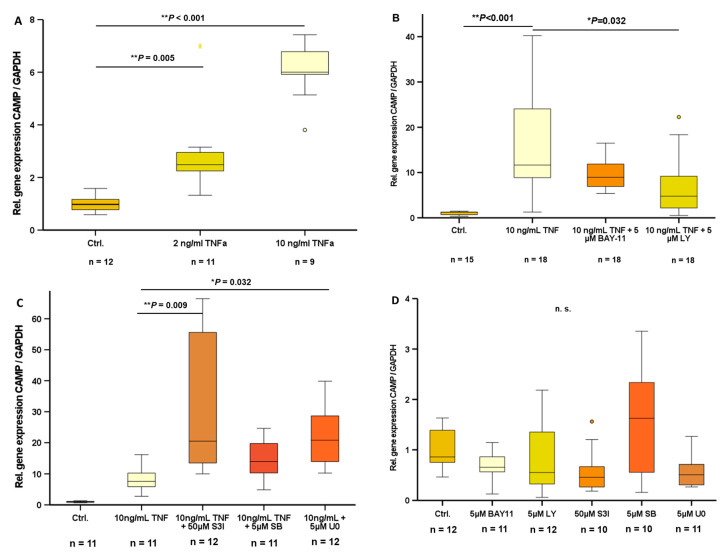
TNFα-induced CAMP gene expression is antagonized by inhibitor of PI3K signaling: TNFα induces CAMP expression in 3T3-L1 adipocytes (**A**). Inhibition of the PI3K pathway by 5 µM LY294002 antagonizes TNFα-induced CAMP expression, whereas inhibition of the NF-κB (5 µM BAY11-7085) (**B**), STAT3 (50 µM S3I-201), MAPK (5 µM SB239063), and MEK-1/-2 (5 µM U0126) pathways does not inhibit TNFα-induced CAMP expression (**C**). Basal CAMP expression is not significantly modified by inhibition of classical signal transduction pathways (**D**) (* *p* < 0.05, ** *p* < 0.01). N = 9–18 samples were investigated. Gene expression levels are given in relative gene expression as compared to control.

**Figure 3 ijms-24-15820-f003:**
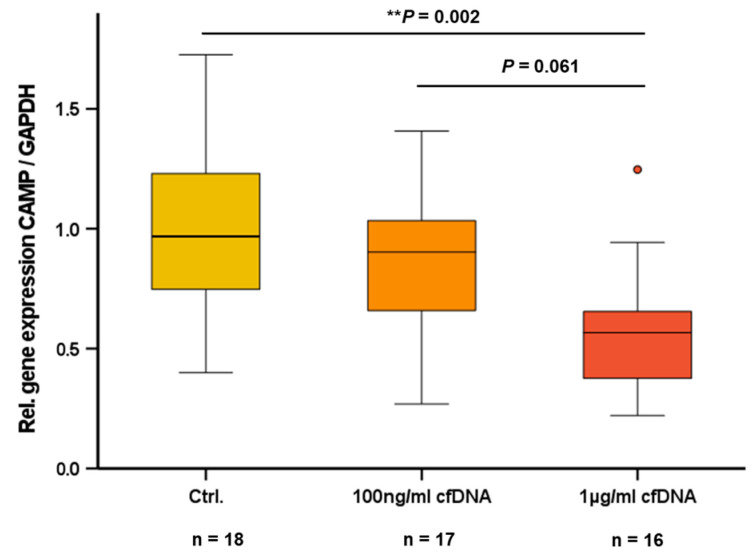
Adipocytic cell-free nucleic acids (cfDNA) impair CAMP expression in 3T3-L1 adipocytes: cfDNA (1 µg/mL) reduces CAMP mRNA levels in 3T3-L1 adipocytes. CAMP expression was investigated by quantitative real-time PCR. Kruskal–Wallis test was applied for calculation of *p* values and statistical significance (*p* < 0.05). N = 16–18 wells were investigated per experimental setting. Gene expression levels are given in relative gene expression as compared to control. (** *p* < 0.01).

**Figure 4 ijms-24-15820-f004:**
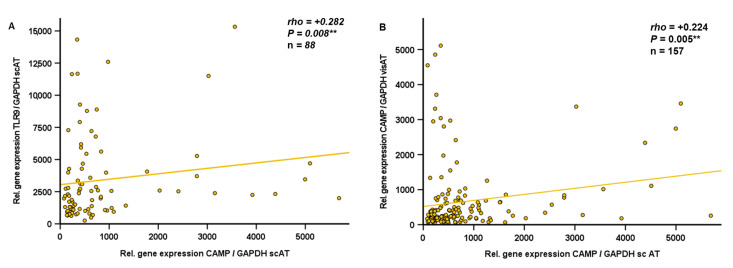
CAMP and TLR9 mRNA levels are positively correlated in human subcutaneous adipose tissue (n = 88) (**A**). CAMP gene expression levels in subcutaneous and visceral adipose tissue are positively correlated (n = 157) (**B**). Specimens of human subcutaneous adipose tissue were collected during bariatric surgery. CAMP and TLR9 expression were investigated by quantitative real-time PCR and Spearman-rho test was applied for calculation of correlation coefficient and statistical significance (** *p* < 0.01).

**Figure 5 ijms-24-15820-f005:**
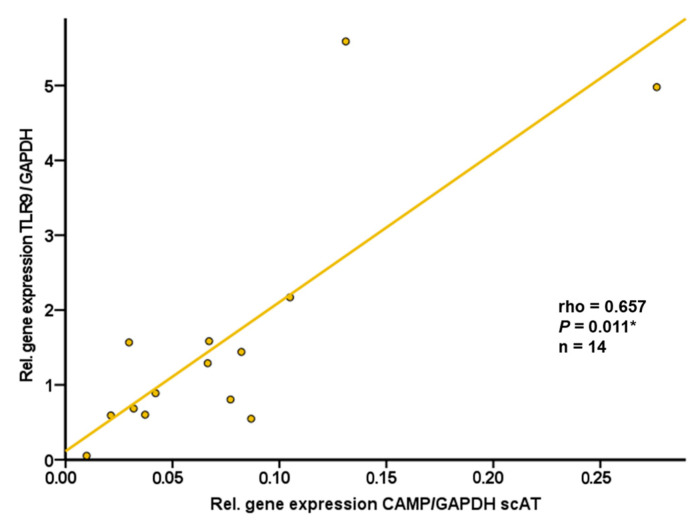
CAMP and TLR9 gene expression levels are positively correlated in murine subcutaneous adipose tissue: TLR9 expression in subcutaneous adipose tissue is positively correlated with CAMP expression in subcutaneous adipose tissue of wildtype mice. Samples of subcutaneous adipose tissue from wildtype mice were collected at the age of 12 months. CAMP and TLR9 expression were investigated by quantitative real-time PCR. The Spearman-rho test was applied for calculation of correlation coefficient and statistical significance (*p* < 0.05). N = 14 samples were investigated. (* *p* < 0.05).

**Figure 6 ijms-24-15820-f006:**
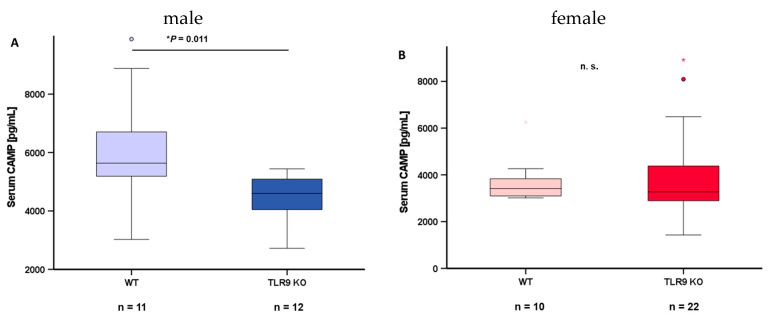
TLR9 KO is associated with reduced systemic CAMP concentrations in male mice: CAMP concentration in blood serum is significantly lower (*p* = 0.011) in male TLR9 KO mice than in wildtype mice (**A**), whereas this difference was not observed in female animals (**B**). Blood serum samples from wildtype and TLR9 KO mice were collected at the age of 5–8 months. Blood serum CAMP concentrations were measured by ELISA and Mann–Whitney U-test was applied for calculation of statistical significance (*p* < 0.05). Samples from n = 10–22 animals per group were investigated. n.s.: not significant, * *p* < 0.05; *: statistical outlier >2.5 interquartile range.

**Figure 7 ijms-24-15820-f007:**
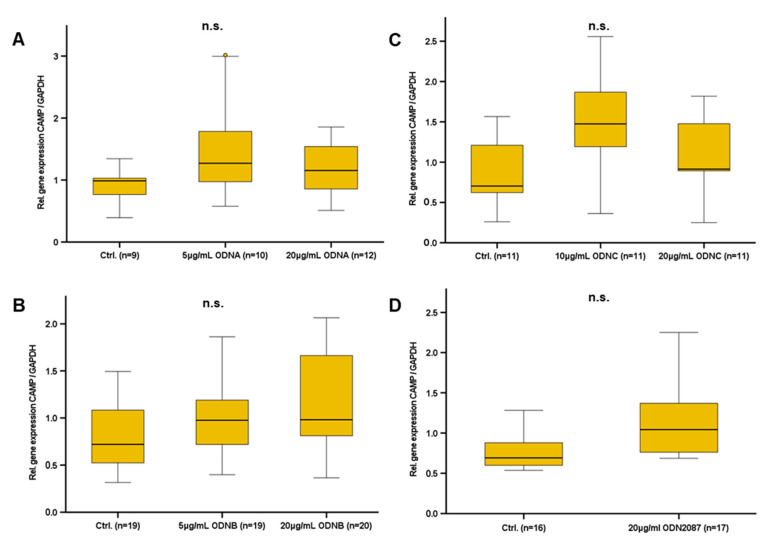
Synthetic TLR9 ligands do not modify CAMP gene expression in murine adipocytes: Stimulation of murine 3T3-L1 adipocytes with various doses of ODNA (**A**), ODNB (**B**), ODNC (**C**), and ODN2087 (**D**) does not significantly affect CAMP gene expression. CAMP expression was investigated by quantitative real-time PCR. The Kruskal–Wallis test (**A**–**C**) and Mann–Whitney U-test (**D**) were applied for calculation of *p* values and statistical significance (*p* < 0.05). N = 11–20 wells were investigated per experimental setting. Gene expression levels are given in relative gene expression as compared to control. n.s.: not significant.

**Figure 8 ijms-24-15820-f008:**
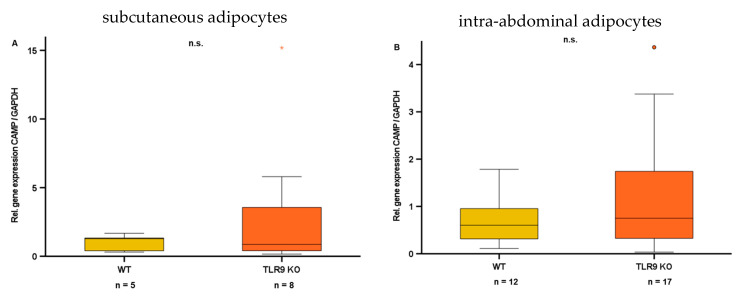
CAMP gene expression in primary adipocytes from subcutaneous (**A**) and intra-abdominal adipose tissue (**B**) does not differ between wildtype (WT) and TLR9 knockout (KO) mice. CAMP expression was investigated by quantitative real-time PCR, and Mann–Whitney U-test was applied for calculation of statistical significance (*p* < 0.05). Samples from n = 5–17 animals per group were analyzed. n.s.: not significant; *: statistical outlier > 2.5 interquartile range.

**Figure 9 ijms-24-15820-f009:**
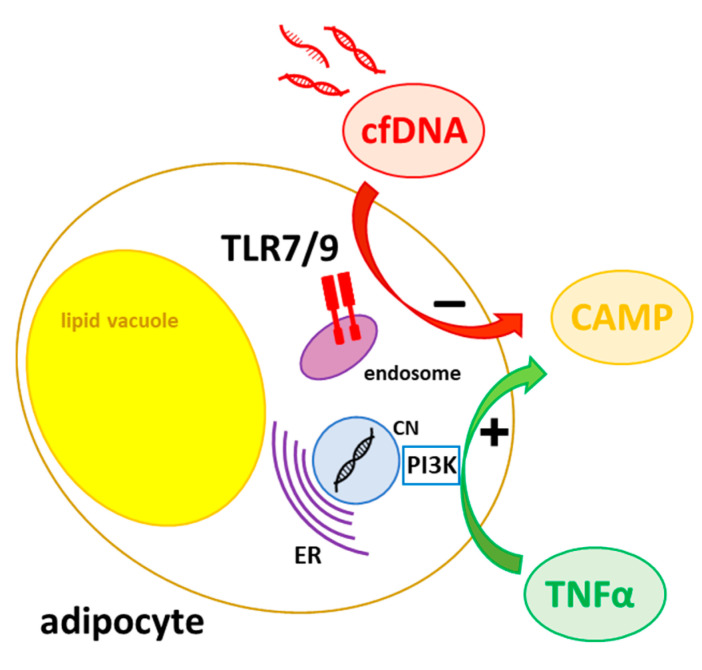
Putative mechanisms of CAMP regulation in adipocytes. Adipocyte CAMP gene expression is induced by TNFα via the PI3K pathway. The impaired CAMP gene expression by cfDNA might be modulated via endosomal receptors, e.g., TLR7/9. ER: endoplasmatic reticulum; CN: cell nucleus.

## Data Availability

The data presented in this study are available on reasonable request from the corresponding author.

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
