# Peer review of "Regulation of Cathelicidin Antimicrobial Peptide (CAMP) Gene Expression by TNFα and cfDNA in Adipocytes"

_ijms, 2023, doi:10.3390/ijms242115820_

Round 1

Reviewer 1 Report

This article is well written.  A few typos should be modified.

1. How about the value of CAMP in Table 1?

2. In the results of 2.2, Author wrote synthetic TLR9.... in murine or human adipocyte, but does not find the results from human adipocyte, if no data on humans, please modified the title of result 2.2.

Author Response

We thank the reviewer for the careful revision of our manuscript and for the positive comments and helpful recommendations. All concerns and corrections could be addressed in the revised version of the manuscript.

  1. How about the value of CAMP in Table 1?

Thank you for this helpful advice. Due to your and another reviewer’s suggestions, we replaced Table 1 by new Figure 2, displaying the relative CAMP values from in vitro experiments in adipocyte cell culture (each in relation to untreated control).

  1. In the results of 2.2, Author wrote synthetic TLR9.... in murine or human adipocyte, but does not find the results from human adipocyte, if no data on humans, please modified the title of result 2.2.

Thank you very much for this hint. We corrected the title of paragraph 2.2.

Reviewer 2 Report

This interesting study shows novel aspects of TLR9-mediated regulation of CAMP in adipocytes and adipose tissues. Results support that TLR9 expression is involved in CAMP regulation in adipocytes in vitro and in adipose tissues in vivo.

However, as presented, the manuscript has several areas for improvement that must be corrected to be considered for publication in IJMS.

-       Fig. 1 must be corrected. Panel labels A and B are missing. Also, it is hard to identify the data of 3T3-L1 adipocytes and 103 in human SGBS cells.

-       In the material and methods section, authors must justify their agonistic oligodeoxynucleotide  ODNA, ODNB, and ODNC variable concentrations.

-       The authors claim that in 3T3-L1 adipocytes, cfDNA dose-dependently reduced CAMP expression with an effective dosage of 1 μg/ml cfDNA (P=0.002). Nonetheless, since the p-value of the results with 100 ng/ml and 1 μg/ml of cfDNA was 0.061, it is not statistically significant. Then, they must mention a trend instead of a dose-dependently expression of CAMP.

-       The assays reported in Figures 3A and 3B must be repeated and re-evaluated. The differences between the assays using the same concentration of TNF-a are a cause for concern. Panel A shows values of CAMP expression above six-fold with 10 ng/ml TNF-a, whereas in panel B the same concentration of TNF-a exerts a CAMP expression near three-fold. The 10 ng/ml TNF-a concentration in panel B lacks significant differences from the control. Then, the results are not consistent with claiming that inhibitors of intracellular signal transduction pathways can antagonize TNF-a-induced CAMP gene expression. Also, the discussion about these results must be revisited.

-       The assays reported in Figure 3C must be re-evaluated. The authors claim that inhibition of STAT3 (S3I-201)-, MAPK (SB239063)-, MEK-1/-2 (U0126)-pathway do not alter TNF-a induced CAMP expression. However, data clearly show a high increase in CAMP expression. Also, the discussion about these results must be revisited.

Author Response

We thank the reviewer for the thorough revision of our manuscript and for the thoughtful recommendations that helped to significantly improve the manuscript’s quality. All concerns and corrections were addressed in the revised version of the manuscript.

  1. Fig. 1 must be corrected. Panel labels A and B are missing. Also, it is hard to identify the data of 3T3-L1 adipocytes and in human SGBS cells.

Thank you for this recommendation. We corrected and elaborated Figure 1 following your recommendation and introduced headlines for improved clarity.

  1. In the material and methods section, authors must justify their agonistic oligodeoxynucleotide ODNA, ODNB, and ODNC variable concentrations.

Thank you for this annotation. Further information on the applied doses of ODNs has been added to the Material and Methods section.

  1. 10, ll. 399-401:

All applied doses were determined following manufacturer’s recommendations for effective dosage as well as by previous preliminary testing for potential cytotoxicity in adipocyte culture.”

  1. The authors claim that in 3T3-L1 adipocytes, cfDNA dose-dependently reduced CAMP expression with an effective dosage of 1 μg/ml cfDNA (P=0.002). Nonetheless, since the p-value of the results with 100 ng/ml and 1 μg/ml of cfDNA was 0.061, it is not statistically significant. Then, they must mention a trend instead of a dose-dependently expression of CAMP.

Thank you for this important comment, we agree and have introduced a correction to the respective paragraph within the Results section.

  1. 4, ll. 142-143:

cfDNA reduced CAMP expression with an effective dosage of 1 µg/mL cfDNA (P=0.002) whereas 100 ng/mL proved to be ineffective (Figure 3).”

  1. The assays reported in Figures 3A and 3B must be repeated and re-evaluated. The differences between the assays using the same concentration of TNF-a are a cause for concern. Panel A shows values of CAMP expression above six-fold with 10 ng/ml TNF-a, whereas in panel B the same concentration of TNF-a exerts a CAMP expression near three-fold. The 10 ng/ml TNF-a concentration in panel B lacks significant differences from the control. Then, the results are not consistent with claiming that inhibitors of intracellular signal transduction pathways can antagonize TNF-a-induced CAMP gene expression. Also, the discussion about these results must be revisited.

Thank you for this valuable comment. Following your recommendation, we added additional in vitro stimulation experiments in order to confirm our findings and to provide improved sample size. The results are displayed by updated Figure 4B indicate a significant inhibition of TNF-induced, substantially elevated CAMP expression by LY294002 but not by BAY11-7085. We introduced this information and the respective changes to the Results and Discussion section of our revised manuscript.

  1. 4-5, ll. 160-166

Inhibition of the PI3K (by 5µM LY294002) signal transduction pathway effectively antagonized TNFα-induced CAMP gene expression (P=0.032) (Figure 4B), whereas inhibition of NF-á´‹B (5µM BAY11-7085), STAT3 (50µM S3I-201), MAPK (5µM SB239063), or MEK-1/2 (5µM U0126) signal transduction pathways did not antagonize TNFα-induced CAMP expression (Figure 4B, C).”

And p.8, ll. 318-321:

This inflammatory elevation of CAMP expression was antagonized by inhibition of PI3K signalling. Of note, pharmacological antagonists of other classical signal transduction pathways exerted either no (NF-á´‹B, MAPK) or even contrary effects (STAT3, MEK-1/2).”

  1. The assays reported in Figure 3C must be re-evaluated. The authors claim that inhibition of STAT3 (S3I-201)-, MAPK (SB239063)-, MEK-1/-2 (U0126)-pathway do not alter TNF-a induced CAMP expression. However, data clearly show a high increase in CAMP expression. Also, the discussion about these results must be revisited.

Thank you for this justified comment. Since we focused on the issue of a potential inhibition of TNF-induced CAMP expression by the tested pathway antagonists, we initially stated in a simplifying manner the absence of any significant effects of S3I-201, SB239063, and U0126. We agree that this can be misinterpreted and have now introduced a more precise description and discussion of the respective data.

  1. 5, ll. 166-168:

In contrast, CAMP mRNA levels were considerably elevated by 50µM S3I-201 and 5µM U0126 treatment, in addition to the TNFα-induced effect (Figure 4C).”

  1. 8, ll. 319-321:

Of note, pharmacological antagonists of other classical signal transduction pathways exerted either no (NF-á´‹B, MAPK) or even contrary effects (STAT3, MEK-1/2).”

Reviewer 3 Report

A significant limitation of the work is the lack of confirmation of the results of PCR analysis using Western blotting. Which would be extremely necessary, since the authors analyze a small number of genes. PCR analysis makes up the bulk of the work and conclusions are drawn based on it. The reliability of the data must be confirmed by additional methods.

1) Images of cell cultures at the stage of active proliferation and after full differentiation are needed.

2) Table 1 is better to arrange in the form of a diagram. The number of repeats for each inhibitor should be clearly stated. N=11-20 – not acceptable 3) Concentrations for all inhibitors (LY294002, BAY11-7085, etc.) should be reported as these compounds may affect different targets depending on the concentration used.

4) The effectiveness of the knockdown must be demonstrated, i.e. change in the expression of the target gene for siRNA. PCR analysis, immunocytochemical staining or Western blotting is required

5) Very interesting studies have been carried out on human tissue and mice. Interesting correlations have been obtained. The description of the results for Figures 6-8 should be more detailed.

6) The authors set one of their goals - to reveal the molecular mechanism. A diagram of the mechanism is needed based on the author's research and, possibly, using literature data.

I am not a native English speaker, but the article is easy to read and I have no serious comments about the quality of the English language

Author Response

We thank the reviewer for the thorough revision of our manuscript and for the helpful recommendations that helped to significantly improve the manuscript’s quality. All expressed concerns and recommended corrections were addressed in the revised version of the manuscript.

  1. A significant limitation of the work is the lack of confirmation of the results of PCR analysis using Western blotting. Which would be extremely necessary, since the authors analyze a small number of genes. PCR analysis makes up the bulk of the work and conclusions are drawn based on it. The reliability of the data must be confirmed by additional methods.

We thank the reviewer for this substantial comment. In the present study we focused on gene expression analysis of TLR9 and CAMP in adipocytes and adipose tissue in order to gain a first insight in putative regulatory relations between both factors of innate immunity. We agree with the reviewer that the focus on mRNA data represents a limitation for the conclusions drawn from the present work. Nonetheless we are convinced that our study provides a valuable data basis for future research to elaborate on, applying further immunological techniques such as Western Blot and immunocytochemistry in order to investigate CAMP regulation in adipocytes and adipose tissue on the protein level.

We address this issue in the revised version of the manuscript.

  1. 9, ll. 370-377:

Overall, we focused on gene expression analysis of TLR9 and CAMP in adipocytes and adipose tissue in order to gain a basic insight in putative regulatory relations be-tween both factors of innate immunity. Predominant analysis on the level of tran-scriptional regulation represents a limitation concerning the physiological conclusions drawn from the present work. Therefore, future research elaborating on the pro-vided data should apply immunological techniques such as Western Blot and im-munocytochemistry in order to analyze CAMP regulation in adipocytes and adipose tissue on the protein level and to confirm and further elucidate the biological relevance of these processes.”

  1. Images of cell cultures at the stage of active proliferation and after full differentiation are needed.

Following your advice, we added new Figure 1A, 1B, 1D and 1E to the revised results section, providing representative light microscopy images of 3T3-L1 and SGBS adipocyte cell culture from different stages of adipocyte differentiation.

  1. Table 1 is better to arrange in the form of a diagram. The number of repeats for each inhibitor should be clearly stated. N=11-20 – not acceptable

Thank you for this helpful advice. Due to your and another reviewer’s suggestions, we replaced Table 1 by new Figure 2, displaying the relative CAMP values from in vitro experiments in adipocyte cell culture (each in relation to the respective, untreated controls) and clearly stating the specific sample size for each experiment.

  1. Concentrations for all inhibitors (LY294002, BAY11-7085, etc.) should be reported as these compounds may affect different targets depending on the concentration used.

Thank you for this annotation. We provide the concentrations of all applied inhibitors in the respective paragraph of the Results and of the Materials and Methods section and also in the updated legend of Figure 4.

  1. 5, ll. 160-168:

Inhibition of the PI3K (by 5µM LY294002) signal transduction pathway effectively antagonized TNFα-induced CAMP gene expression (P=0.032) (Figure 4B), whereas inhibition of NF-á´‹B (5µM BAY11-7085), STAT3 (50µM S3I-201), MAPK (5µM SB239063), or MEK-1/2 (5µM U0126) signal transduction pathways did not antagonize TNFα-induced CAMP expression (Figure 4B, C). In contrast, CAMP mRNA levels were considerably elevated by 50µM S3I-201 and 5µM U0126 treatment, in addition to the TNFα-induced effect (Figure 4C).”

  1. 10, ll. 404-409:

Furthermore, co-stimulation experiments were performed with 10 ng/mL TNFα and inhibitors of different signal transduction pathways (NF-á´‹B inhibitor BAY-11 (5 mM), STAT3 inhibitor S3I-201 (50 mM), selective MAPK inhibitor SB239063 (5 mM), MEK-1/-2 inhibitor U0126 (5 mM), and phosphatidylinositol 3-kinase (PI3K) inhibitor LY294002 (5 mM), all purchased from Merck). Dosage was applied as described in our previous study.”

  1. The effectiveness of the knockdown must be demonstrated, i.e. change in the expression of the target gene for siRNA. PCR analysis, immunocytochemical staining or Western blotting is required

We agree that this is indeed a crucial issue not being sufficiently addressed in the original version of the manuscript. We verified the effectiveness of siRNA-mediated knockdown by quantification of intracellular TLR9 gene expression. As is displayed in new Figure 9 within the Materials and Methods section, TLR9 mRNA levels in mature adipocytes were specifically reduced by more than 50 % by continuous transfection with TLR9 siRNA throughout adipocyte differentiation when compared to control cells receiving non-target siRNA.

  1. Very interesting studies have been carried out on human tissue and mice. Interesting correlations have been obtained. The description of the results for Figures 6-8 should be more detailed.

Thank you for this positive comment. As suggested, we improved the presentation and discussion of the correlation analysis of in vivo data in the revised manuscript.

  1. 6, ll. 209-213:

CAMP serum concentrations were significantly lower (P=0.011) in male TLR9 KO mice when compared to wildtype mice (WT: 6122.70 ± 1962.79 pg/mL vs. TLR9 KO: 4445.01 ± 826.42 pg/mL) (Figure 6A). This difference between genotypes was not observed in female mice (WT: 3735.38 ± 973.51 pg/mL vs. TLR9 KO: 3937.66 ± 1847.37 pg/mL) (Figure 6B).”

  1. The authors set one of their goals - to reveal the molecular mechanism. A diagram of the mechanism is needed based on the author's research and, possibly, using literature data.

Following your kind advice, we added a schematic diagram as new Figure 10 to the Discussion section of the revised manuscript, proposing a mechanistical model based on our new data.

  1. 13, ll. 545-549:

“Figure 10: Diagram of proposed mechanisms of CAMP regulation in adipocytes.

In adipocytes, CAMP gene expression is induced by TNFα via PI3K pathway and impaired bycfDNA. Of note, siRNA mediated knockdown of TLR9 is associated with suppressed CAMP gene expression, potentially associated to the inhibition of adipocytic differentiation.  ER: endoplasmatic reticulum; CN: cell nucleus”

Round 2

Reviewer 2 Report

The manuscript was rightly corrected, and the new version is significantly improved.

Author Response

We want to thank the reviewer again for the careful revision of our manuscript and for the helpful recommendations. Thanks to your excellent suggestions, we feel that we were able to significantly improve our manuscript. 

Reviewer 3 Report

My opinion remains the same. Without the use of additional methods, the conclusions in this work remain unconfirmed. A decrease in gene expression after knockdown by 50% is not an indicator of the high efficiency of knockdown. Western blot analysis required.

Author Response

We thank the reviewer for the careful revision of our manuscript, and we highly appreciate your comment. In comment 5 of the initial review process, reviewer 3 asked for verification of siRNA-mediated knock-down of TLR9 in adipocytes by PCR analysis, immunocytochemical staining or Western blotting; in our revised manuscript, we chose to verify knock-down by PCR analysis. In the new figure 9 of our revised manuscript, we were able to demonstrate significantly reduced TLR9 mRNA levels (p=0.016; TLR9 mRNA levels were reduced by approximately 53%; 28.8 ± 5.0 versus 13.5 ± 3.4 (values representing TLR9 mRNA expression relative to GAPDH ((TLR9/GAPDH)*106), mean +/- SEM)). This TLR9 mRNA knockdown resulted in a highly significant decrease of CAMP gene expression (p<0.001, figure 5). Additionally, we investigated CAMP serum concentrations in TLR9 knockout mice in vivo. Congruent with the findings in vitro, CAMP serum concentrations are significantly lower in TLR9 knockout mice.

Nonetheless, we completely agree with the reviewer that reduced TLR9 mRNA levels are an indirect measure; unfortunately, at least in our hands detecting TLR9 protein by Western blotting is unreliable, an observation that has been made by other groups as well (personal communication). Additionally, a reduced amount of TLR9 protein levels might still be sufficient for TLR9 signaling. Thus, to our mind future experiments should definitely elaborate on our results: In particular, future experiments should investigate CAMP gene expression in primary murine adipocytes isolated from male versus female TLR9 knockout mice to verify our observations reported in adipocytes treated by siRNA-mediated knockdown of TLR9 expression and to further elaborate on a potential sexual dimorphism and on a potential impact of impaired adipocyte differentiation.

We added an additional comment in the discussion section of our manuscript (p. 9, ll. 349-354):

“In particular, future experiments should investigate CAMP gene expression in primary murine adipocytes isolated from male versus female TLR9 knockout mice to verify the observations reported in adipocytes treated by siRNA-mediated knockdown of TLR9 expression and to further elaborate on a potential sexual dimorphism and on a potential impact of impaired adipocyte differentiation”.

Round 3

Reviewer 3 Report

Since the study is based on gene knockout, and the level of the target protein is not shown, then all the results remain debatable for me. By all the rules for conducting such studies, the assessment of protein levels is strictly necessary.

Author Response

We thank the reviewer for the careful revision of our manuscript, and we highly appreciate your comment. As the reviewer’s comment was related to the TLR9 knockdown via siRNA, we completely agree with the reviewer that reduced TLR9 mRNA levels are an indirect measure; unfortunately, at least in our hands detecting TLR9 protein by Western blotting is unreliable. To ensure a sufficient TLR9 knockout, we elaborated our study by investigating levels of CAMP gene expression in murine primary adipocytes of TLR9 knockout mice and wildtype mice (new Figure 8, p.7, ll. 271-279). In this mouse model, CAMP gene expression was not altered significantly by TLR9 knockout. Thus, we can conclude that the inhibitory effect of cfDNA on CAMP expression we observed in adipocytes is mediated via additional receptors, for example TLR7, as cfDNA is a known TLR7 agonist as well. Future studies are necessary to explore the underlying mechanisms of impaired CAMP gene expression by cfDNA in adipocytes.

We added the new experiments in the manuscript and the discussion section was revised. (p. 9, ll. 364-377):

“Additionally, we analyzed CAMP gene expression in primary adipocytes from wildtype and TLR9 knockout mice. Importantly, we observed no significant difference between CAMP gene expression levels in primary adipocytes from wildtype mice as compared to TLR9 knockout mice, neither for subcutaneous nor intra-abdominal adipose tissue. Together, these findings suggest that - although cfDNA impairs CAMP gene expression in adipocytes in vitro - the association of CAMP and TLR9 expression levels in adipose tissues presumably mainly depends on other cell types different from adipocytes, like monocytes or macrophages in the stromal vascular fraction of adipose tissue, which are a known source of CAMP [37]. Furthermore, the observed inhibitory effect of cfDNA on CAMP gene expression in adipocytes might be mediated via additional receptors, for example TLR7, as cfDNA is a putative TLR7 agonist as well [34]. Future studies on putative regulatory interactions of CAMP and TLR7 in adipocytes are necessary to verify this hypothesis and to further elucidate the mechanisms un-derlying CAMP expression in adipose tissue.”.

  1. 7, II. 257-269:

2.8. Knockout of TLR9 does not affect CAMP gene expression in murine primary subcutaneous and intra-abdominal adipocytes

We detected significant correlations of CAMP and TLR9 gene expression in hu-man and murine subcutaneous adipose tissue, and TLR9 knockout in male mice resulted in reduced CAMP serum levels. However, synthetic TLR9 ligands – as opposed to cfDNA – did not modify CAMP gene expression in murine adipocytes in vitro. In order to further investigate the impact of TLR9 on CAMP expression in adipocytes, we therefore tested the cellular effect of TLR9 knockout on CAMP gene expression in primary adipocytes isolated from wildtype mice and TLR9 KO mice. Due to the diverging constitution of subcutaneous and intra-abdominal adipose tissue, we evaluated adipocytes from both compartments separately (Figure 8A, B). Of note, we detected no difference in CAMP gene expression in subcutaneous (Figure 8A) and intra-abdominal (Figure 8B) adipocytes of wildtype and TLR9 knockout mice.”

Round 4

Reviewer 3 Report

The authors have seriously revised the article in the light of my main comment. Since we all face the problem when it is not possible to apply the method that the reviewer requires of us, I consider the data presented to be quite conclusive. The discussion of the results has been expanded, making the article easier to understand.

The authors should discuss the relationship of protein kinases, especially PI3K, with Ca2+ signaling and regulation of adipocyte physiology. https://pubmed.ncbi.nlm.nih.gov/34065973/